# Mint Plants (*Mentha*) as a Promising Source of Biologically Active Substances to Combat Hidden Hunger

Taras Hutsol [1,*], Olesia Priss [2,*], Liudmyla Kiurcheva [2], Maryna Serdiuk [2], Katarzyna Panasiewicz [3], Monika Jakubus [4], Wieslaw Barabasz [5], Karolina Furyk-Grabowska [6] and Mykola Kukharets [7]

1   Department of Mechanics and Agroecosystems Engineering, Polissia National University, 10-008 Zhytomyr, Ukraine

2   Department of Food Technologies and Hotel and Restaurant Business, Dmytro Motornyi Tavria State Agrotechnological University, 72-312 Melitopol, Ukraine; liudmyla.kiurcheva@tsatu.edu.ua (L.K.); maryna.serdiuk@tsatu.edu.ua (M.S.)

3   Department of Agronomy, Poznan University of Life Sciences, Dojazd 11, 60-632 Poznań, Poland; katarzyna.panasiewicz@up.poznan.pl

4   Department of Soil Science and Microbiology, Poznan University of Life Sciences, Szydłowska 50, 60-656 Poznań, Poland; monika.jakubus@up.poznan.pl

5   Faculty of Technical Sciences and Design Arts, National Academy of Applied Sciences in Przemyśl, Książąt Lubomirskich 6, 37-700 Przemyśl, Poland; rrbaraba@cyf-kr.edu.pl

6   Department of Production Engineering, Logistics and Applied Computer Science, Faculty of Production and Power Engineering, University of Agriculture in Kraków, Balicka 116B, 30-149 Kraków, Poland; karolina.furyk-grabowska@urk.edu.pl

7   Department of Electrification, Production Automation and Engineering Ecology, Polissia National University, 10-008 Zhytomyr, Ukraine; da-h@pdatu.edu.ua

*   Correspondence: wte.inter@gmail.com (T.H.); olesia.priss@tsatu.edu.ua (O.P.)

**Abstract:** Hidden hunger, also known as micronutrient deficiency, is a form of undernutrition, which is exacerbated when food security is fragile. However, the amount of phytonutrients in the diet can be increased by using underutilized species, such as fresh mint greens (*Mentha*). These plants have a high biological value due to the high content of biologically active substances. Plants of the genus *Mentha*, however, differ significantly in their chemical composition and, thus, nutritional value. The main objective of this study was to evaluate the content of phytonutrients in different Ukrainian species of mint: peppermint (*Mentha piperita*), horsemint (*Mentha longifolia* L.), silver-leaved horsemint (*Mentha longifolia S.*), and spearmint (*Mentha spicata* L.), and to determine their stability after drying and freezing. After studying the chemical composition of fresh, dried, and frozen plants of these species, it was established that Ukrainian mint species offer a robust set of phytonutrients and can be used as ingredients of the so-called "functional foods." The biologically active substances in mint are concentrated during drying. Moreover, such raw materials are easily stored and used as an ingredient. However, the losses of ascorbic acid due to drying mint reach 70%; of carotenoids—approximately 10–15%; of chlorophylls—21–38%; and of phenols—19–29%. Peppermint and spearmint were observed to have higher stability of biologically active compounds. As for the freezing, the best stability of the chemical composition was demonstrated by field mint and spearmint samples. Therefore, a targeted selection of mint types and varieties prior to processing will allow preserving maximal preservation of a maximum amount of biologically active substances, increasing the content of phytonutrients in finished products, and preventing the development of hidden hunger.

**Keywords:** mint greenery; biologically active substances; total dry matter; sugars; titratable acidity; ascorbic acid; carotenoids; chlorophyll; polyphenols; hidden hunger

## 1. Introduction

The global food system currently faces acute problems caused by the coronavirus pandemic and the Russian aggression against Ukraine. War significantly increased the negative trends of the world's food system and violated the guaranteed obligations to ensure its security [1]. As the prolonged effects of these crises are looming, overlapping with the immediate problem of food supply, the problem of hidden hunger, that is, the lack of important phytonutrients, will deepen and have a strong impact on the health of future generations [2]. The food system must use all possible ways to ensure its sustainability and overcome the negative consequences of crises.

One of the ways to increase the sustainability of food systems is the use of neglected and underutilized species. Today, it is estimated that only a quarter of approximately 600 types of vegetable crops are used for food [3]. Regular consumption of underutilized vegetables is an effective way to fight hidden hunger, maintaining a varied and healthy diet.

Biological activity of plant metabolites

Primary and secondary metabolites of plants have high biological activity. Many raw plant compounds, such as pigments (chlorophylls [4,5] and carotenoids [6]), polyphenols [7], and ascorbic acid [8], have antioxidant and antiradical properties. Carbohydrates are primary metabolites necessary for the development and maturation of plant tissues. Studies have shown that they play an important role in maintaining normal levels of reactive oxygen species and are positioned as antioxidants [9]. On the other hand, phytonutrients activate immunity and stimulate the body's reserve defense mechanisms by directly affecting metabolism, increasing personal performance, and preventing stress-related problems and other adverse factors [10–13].

The possibility of integrated use of plant components in food products to give them multifunctional properties is of particular relevance [14–16]. Adjustments are made to the chemical composition of food with non-traditional raw plant components, which are a source of nutrients and biologically active substances, and have a pronounced physiological effect and multifunctional action (antioxidant, antiseptic, immunomodulatory, radioprotective, anticancer, etc.) [15–17]. Many families of plants like Asteraceae (e.g., *Artemisia campestris* L.), Zingiberaceae (e.g., *Alpinia galangal* (L.) Willd.), Caryophyllaceae (e.g., *Corrigiola litoralis* subsp. foliosa (Pérez Lara) Devesa), etc., have been consumed since ancient times without the knowledge of nutraceuticals due to their positive impact on health, caused by the particularly high content of secondary metabolites [18]. Lamiaceae, or the mint family, is another family which is rich in medicinal plants. It contains more than 3000 species that have a cosmopolitan distribution [19]. Numerous members of Lamiaceae are cultivated not only for the decorative foliage, but rather on the industrial scale, since they are of significant value for pharmacy, perfumery, cosmetology, and culinary, as flavorings and sources of functional food.

Selecting Ukrainian raw plant material for food enhancement

The growing trend in the implementation and effectiveness of natural aromatics, flavoring, and essential oil raw plants is a significant motivation to study the biological and nutritional value of mint. To develop this segment of the raw plant market, it is of interest to expand the range and species composition of this raw ingredient.

The natural flora of Ukraine is an inexhaustible source of promising raw plant materials for food enrichment [20–22]. Due to purposeful scientific research, a wide range of raw plants is used in the food industry and medicine [23,24]. The introduction of highly active functional ingredients in food recipes makes the therapeutic effectiveness of their impact on the human body compatible with classical therapeutic agents. Although these products are not medicines, they are included in the range of natural methods for the preservation and improvement of health [25].

The selection of raw plant materials in our region reveals a great variety of unique species of wild, alien, and cultivated plants [26,27]. Green crops, grown in open and sheltered soil, have a complex of phytonutrients and can be a food ingredient [28–30].

Within this framework, plants of the Lamiaceae family (*Lamiaceae Lindl.*), the most common both in Ukraine and in the global flora, represented by mint (*Mentha* L.), are of great interest. The genus Mentha has 18 to 30 species, which differ significantly in chemical composition [31–34]. However, not all species and varieties of mint are equally valuable [35].

Value of mint as an aromatic and medicinal plant

The value of mint as a spicy, aromatic, and medicinal plant is very high. Medicines produced from mint have a calming, antispasmodic, antiseptic, and analgetic effect [36–38]. The leaves or the upper part of plants are used as phyto-raw materials for food production. Fresh mint greens have high biological value due to the complex of biologically active substances: alkaloids, saponins, organic acids, vitamins, carotenoids, chlorophylls, and macro- and micronutrients [39,40]. In the complex of biologically active substances of mint, phenols take precedence, being represented by phenolic acids (chlorogenic, caffeic, rosemary, ferulic), flavonoids (luteolin and its glycosides), and tannins [41,42]. Caffeic, rosemary, and ferulic acids were identified in mint [42,43]. Other authors note the significant antibacterial and fungicidal activity of mint on a wide range of pathogenic microorganisms and fungi [44–47].

The vast majority of the research regarding the biochemistry of mint is focused on the content and qualitative indicators of essential oil in it, since mint is widely used in the cosmetic industry and in culinary flavor dishes and drinks. The content of essential oil in different types of mint varies significantly. Brazilian researchers found that among plants of the genus *Mentha piperita* L., Chocolate mint shows the highest content of essential oil (0.53%), whereas, among plants of the genus *Mentha spicata L.*, content is the highest for the Homegrown mint (0.17%). The lowest essential oil content (0.05%) was recorded in Himalayan silver mint (*Mentha longifolia* (L.) *Huds*) [48]. Moreover, the composition of the essential oil itself differs in varieties of the same plant species. In *M. piperita* (chocolate mint), the prevailing compounds are menthofuran (23.70%), menthone (17.27%), d-neoisomenthol (14.35%), and pulegone (10.74%). In M. piperita (grapefruit mint), however, the linalool (25.43%) and linalyl acetate (51.35%) are of the most importance [48]. Mimica-Dukić et al. studied the antimicrobial activity and free radical scavenging capacity (RSC) of essential oils from *Mentha aquatica* L., *Mentha longifolia* L., *and Mentha piperita* L. [49]. They showed that essential oils from all types of mint have a very strong antibacterial effect, in particular against strains of *Escherichia coli. M. piperita* essential oil was the most efficient; in addition, all studied oils showed significant fungistatic and fungicidal activity. Tafrihi et al. emphasized anticancer activities that complement the antimicrobial role of mint-derived compounds [50].

For the fortification of food products, however, it is practical to use the whole plant. Therefore, more attention should be paid to the study of the complete biochemical composition, which can provide knowledge regarding the content of both beneficial and problematic compounds. For example, organic acids, such as citric and valeric, also showed strong antimicrobial activities [51]. On the other hand, the sugar/acid ratio (defined as the ratio of sugar content to titrated acidity) is particularly important for the food industry, since it can significantly affect the organoleptic indicators of ready products. To date, little is known about the presence of dietary fibers, mineral compounds, carbohydrates, and organic acids in mint plants.

Mint cultivation is seasonal, and the plant itself is short-lived, which hinders the supply of the required number of raw materials to the food industry. Therefore, many scientific works are devoted to the search for processing methods for a prolonged postharvest shelf life, which would contribute to the maximum preservation of valuable biologically active substances in raw material.

The qualitative and quantitative composition of fresh, as well as dried or frozen mint, has significant differences [52,53], which should be taken into account when using it in food production. Fresh mint has a limited shelf life, even under refrigeration conditions [54], so the question of the possibility of its preservation is relevant.

The quality of the final product depends on the correct choice of the raw material processing method, as some chemical compounds can change their bioavailability under the influence of technological factors, reduce, and sometimes even lose their functional activity.

The research goal

The main objective of this study was to evaluate the content of some phytonutrients in different Ukrainian types of mint and to determine their stability after drying and freezing.

To achieve the research goal, the following tasks were determined:

- To screen the local phyto-raw plants as a source of biologically active substances to be used in food recipes;
- To analyze the content of biologically active substances in the samples of the mint of the most common species and its modification depending on the method of processing;
- To justify the choice of the species of mint and the method of their processing to maximize the quality of the finished product.

## 2. Materials and Methods

### 2.1. Materials

The subject of the research was freshly cut, dried, and frozen samples of four mint species: peppermint (*Mentha piperita*), horsemint (*Mentha longifolia* L.), silver-leaved horsemint (*Mentha longifolia silver*), and spearmint (*Mentha spicata* L.) (Figure 1).

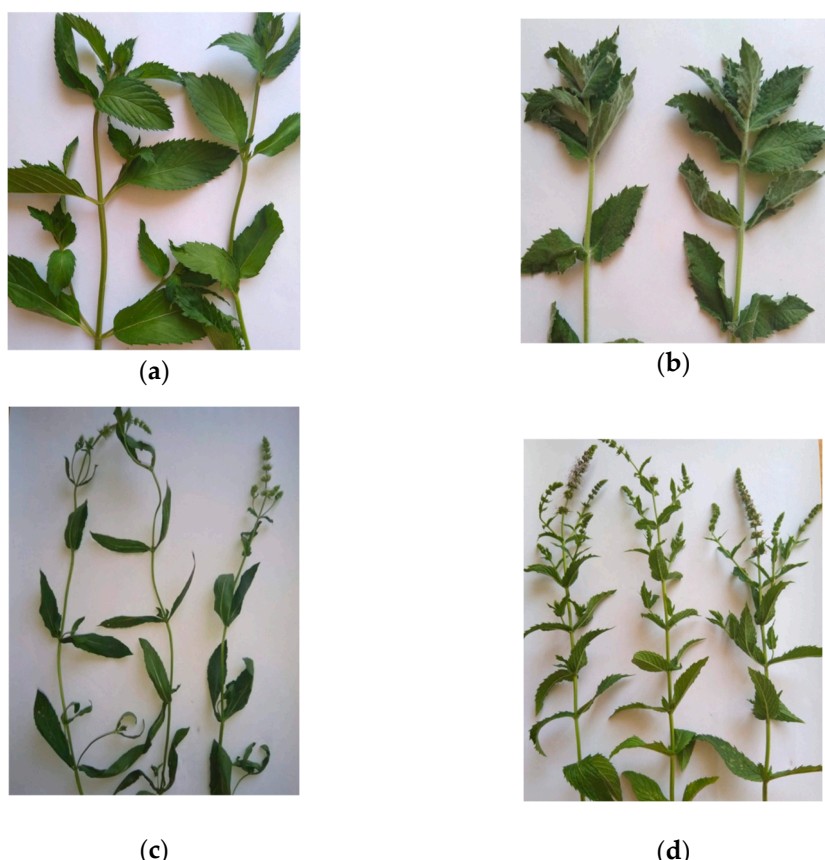

(**a**)  (**b**)

(**c**)  (**d**)

**Figure 1.** Species of Mint: (**a**)—Peppermint (*Mentha piperita*), (**b**)—Silver-leaved Horsemint (*Mentha longifolia silver*), (**c**)—Spearmint (*Mentha spicata* L.), and (**d**)—Horsemint (*Mentha longifolia* L.). Source: author's photos of experimental plants.

The research material was the aerial part of wild mint plants growing in the soil and climatic conditions of the Zaporizhzhia region (46.821124, 35.264201), Ukraine (base farm of Dmytro Motornyi Tavria State Agrotechnological University). Fresh, young, and healthy

plants with a characteristic shape and color, not yellowed and not withered, without sunburn, soil pollution, and excess external moisture were selected. Mint was harvested with stems and leaves for botanical diversity, but without roots. The plants were collected during the flowering period, in the morning, when the content of essential oil in the leaves was maximum [55,56].

Uncrushed mint leaves were dried using convection (T = 50 °C). Before freezing, the mint was chopped with a steel knife into 1 cm long pieces and packed in 500 g bags of 0.05 mm thick plastic film. The freezing temperature was −30 °C, and the frozen greens were stored at −18 °C.

In the course of the research, changes in the chemical composition depending on the type of plants and the method of processing were analyzed. The chemical composition of fresh, dried, and frozen green mint was determined by the following indicators: total dry matter, sugars, titratable acidity, ascorbic acid, phenolic substances, carotenoids, and chlorophyll.

### 2.2. Determination of Dry Matter and Total Sugar Content

The proportion of dry matter was determined in the early budding phase using the thermogravimetric method. The mass fraction of the sugars was determined using the ferricyanide method in accordance with DSTU (National Standard of Ukraine) 4954 [57], p. 108. The method is based on the oxidation of reducing sugars with an alkaline solution of potassium ferricyanide, followed by measuring the intensity of the oxidant coloration using a photoelectrocolorimeter.

During the experiment, two filtrates were prepared. The first filtrate (F1) was prepared from 50 g of plant material, and transferred with water into a 250 mL flask to approximately half of the flask volume. A total of 1 mL of 15% sodium carbonate solution was added for neutralization. The content of the flask was kept in a water bath at T = 80 °C for 20 min, with periodical shaking. After cooling, 15 mL of 14.5% zinc sulfate and 1 mL of 4% sodium hydroxide solutions were added to the flask. The flask content was then shaken, diluted to 250 mL with distilled water, and filtered through a paper filter. The second filtrate (F2) was prepared by the two-fold dilution of 50 mL of the first filtrate.

The oxidizing reagent was prepared in a heat-resistant flask by mixing 5 mL of 2.5 N sodium hydroxide, 20 mL of potassium ferricyanide solution, and 2 mL of water, and further applied to 8 mL of F2. The contents of the flask were boiled on the stove for 1 min, cooled, and filtered. Then, the optical density was measured at a wavelength of 440 nm.

Next, sucrose inversion was performed. Undiluted filtrate (F1) in the amount of 50 mL was added to a 100 mL flask. A total of 5 mL of concentrated hydrochloric acid was added. The flask was heated in a water bath at 70 °C for 5 min, and then cooled. The solution was neutralized with sodium hydroxide, and orange methyl was used as an indicator. The volume of the solution was brought to 100 mL with water. The preparation of the oxidizing reagent for F1 and optical density measurements were performed in the same way as for the F2. The mass of reduced saccharides was determined in milligrams according to the obtained values of optical density using the calibration graph. The calibration graph was based on sucrose dilutions.

### 2.3. Determination of the Mass Fraction of Titrated Acidity

The titrated acidity was determined according to DSTU 4957 (National Standard of Ukraine) [57], p. 75. The mass fraction of titrated acidity was expressed in terms of predominant (malic) acid.

### 2.4. Determination of Chlorophylls and Carotenoids

The number of chlorophylls (a + b) and carotenoids was determined by the optical density of the fraction extracted with pure acetone. The optical density measurements were performed spectrophotometrically at wavelengths 440.5, 644, and 662 nm [58]. The method is described in detail above [28].

### 2.5. Determination of the Total Phenolic Content

The number of polyphenolic substances was determined as previously described [28]. Briefly, the method comprises the complexation reaction of polyphenols with a Folin–Denis reagent and the formation of colored substances, followed by the determination of the optical density of the solution. Measurements were conducted according to DSTU (National Standard of Ukraine) 4373: 2005 [57], p. 275.

### 2.6. Determination of Ascorbic Acid Content

Determination of ascorbic acid content was determined, as described earlier [28]. Briefly, an iodometric measurement of the reducing ability of ascorbic acid was performed [59]. During the reaction, ascorbic acid is oxidized along with the restoration of the free iodine from the potassium iodide. The amount of the generated free iodine was determined based on the color reaction using starch.

All experiments were performed in triplicates, and data were presented as mean $\pm$ standard deviation. Data analysis was performed by an analysis of variance, with mean separation by LSD at the 0.05 level.

## 3. Results

Maximum levels of dry matter, sugars, and titrated acids and pigments were observed in the greens of spearmint and peppermint. Horsemint and silver-leaved horsemint (11.15 and 10.56 mg $\times$ 100 g$^{-1}$, respectively) had higher levels of ascorbic acid than other specimens. The highest number of phenolic substances was found in the greens of silver-leaved horsemint (138.76 mg $\times$ 100 g$^{-1}$), while the lowest one was observed in horsemint (122.71 mg $\times$ 100 g$^{-1}$). Detailed results for chemical composition of fresh, dried, and frozen mint analyzed in the course of the research are reported in Table 1.

**Table 1.** Chemical composition of mint greens, M $\pm$ SD, n = 3.

| Species of Mint | Sugars, g $\times$ 100 g$^{-1}$ | Titratable Acidity, mg $\times$ 100 g$^{-1}$ | Ascorbic Acid, mg $\times$ 100 g$^{-1}$ | Carotenoids, mg $\times$ 100 g$^{-1}$ | Chlorophyll (a + b), mg $\times$ 100 g$^{-1}$ | Total Phenolic Content, mg $\times$ 100 g$^{-1}$ | Dry Matter, % |
|---|---|---|---|---|---|---|---|
| | | | Fresh mint | | | | |
| Peppermint | 2.26 ± 0.10 | 0.54 ± 0.010 | 7.63 ± 0.42 | 21.15 ± 0.32 | 120.72 ± 0.78 | 135.03 ± 1.23 | 18.25 ± 0.65 |
| Horsemint | 2.02 ± 0.09 | 0.39 ± 0.017 | 11.15 ± 0.55 | 19.68 ± 0.24 | 100.62 ± 0.85 | 122.71 ± 0.86 | 17.20 ± 0.56 |
| Silver-leaved horsemint | 1.88 ± 0.12 | 0.45 ± 0.015 | 10.56 ± 0.27 | 16.29 ± 0.26 | 82.10 ± 0.84 | 138.76 ± 1.15 | 16.62 ± 0.43 |
| Spearmint | 2.59 ± 0.16 | 0.61 ± 0.011 | 9.39 ± 0.67 | 23.32 ± 0.38 | 148.11 ± 1.01 | 135.04 ± 1.12 | 18.50 ± 0.64 |
| LCD$_{05}$ | 0.27 | 0.03 | 1.12 | 0.06 | 1.56 | 1.53 | 1.02 |
| | | | Dried mint | | | | |
| Peppermint | 11.07 ± 0.14 | 1.85 ± 0.058 | 11.10 ± 0.56 | 97.30 ± 1.21 | 450.15 ± 1.73 | 227.80 ± 1.54 | 92.32 ± 0.36 |
| Horsemint | 10.32 ± 0.08 | 1.64 ± 0.065 | 17.16 ± 0.36 | 89.08 ± 1.17 | 385.60 ± 1.86 | 160.20 ± 1.34 | 91.16 ± 0.54 |
| Silver-leaved horsemint | 9.80 ± 0.14 | 1.77 ± 0.069 | 16.82 ± 0.48 | 78.03 ± 1.27 | 280.16 ± 2.01 | 221.92 ± 1.24 | 90.84 ± 0.31 |
| Spearmint | 12.69 ± 0.23 | 2.15 ± 0.012 | 13.88 ± 0.45 | 107.6 ± 1.36 | 583.70 ± 2.45 | 202.18 ± 1.36 | 92.98 ± 0.64 |
| LCD$_{05}$ | 0.36 | 0.12 | 0.84 | 2.36 | 4.23 | 2.79 | 1.05 |
| | | | Frozen mint | | | | |
| Peppermint | 2.13 ± 0.15 | 0.50 ± 0.012 | 7.39 ± 0.21 | 20.71 ± 0.41 | 118.08 ± 1.26 | 123.69 ± 0.94 | 18.24 ± 0.37 |
| Horsemint | 1.83 ± 0.12 | 0.32 ± 0.021 | 10.77 ± 0.34 | 19.27 ± 0.32 | 98.06 ± 0.87 | 116.51 ± 1.05 | 17.20 ± 0.51 |
| Silver-leaved horsemint | 1.72 ± 0.17 | 0.38 ± 0.010 | 10.12 ± 0.61 | 15.84 ± 0.38 | 79.83 ± 0.56 | 131.69 ± 1.23 | 16.63 ± 0.41 |
| Spearmint | 2.34 ± 0.24 | 0.55 ± 0.013 | 8.93 ± 0.43 | 22.76 ± 0.17 | 143.51 ± 0.74 | 126.92 ± 1.17 | 18.50 ± 0.54 |
| LCD$_{05}$ | 0.15 | 0.03 | 0.86 | 0.61 | 0.86 | 1.27 | 0.80 |

Source: experimental data obtained by the authors.

## 4. Discussion

The biosynthesis of primary and secondary metabolites in plants depends on environmental conditions, genotype, stage of plant development, soil composition, and other abiotic and biotic factors [60–63], as confirmed by this research. Samples of local peppermint contained more sugar than samples from Lithuania and Poland [64]. This can be explained by the strong influence of the humidity and heat resources on the plant fund of sugars. The region of Zaporizhia is situated in the subzone of the Southern Steppe of Ukraine. The district is characterized by a high level of heat supply (the annual sum of temperatures above 10 °C is 3400–3600) and the lowest level of humidification in the country (annual hydrothermal coefficient of 0.5–0.7 units), which is characteristic of a zone of significant drought. Hence, plants in this zone accumulate a significant number of soluble sugars to regulate osmotic activity. This ensures the protection of cellular structures from elevated temperatures and the maintenance of the water-cellular balance and membrane stability [65].

On the contrary, the content of carotenoids and chlorophylls in peppermint was much lower than described by the Lithuanian authors, being comparable with the data from Iraq scientists [66]. Once again, high temperatures during plant cultivation led to the decreased efficiency of photosynthesis and, therefore, lower chlorophyll content. When assessing the presented data variation, one must also consider whether pigment content is calculated related to the dry or fresh weight (in this work, we provide data for the fresh weight). We also observed a lower ascorbic acid content than reported by other authors who studied Polish peppermint [67]. This observation correlates with the known tendency to increase ascorbic acid content while shifting the zone of plant cultivation toward the North [68].

To sum up, we observed significant differences for all the indicators investigated, which is probably explained by differences in the plant genotype. These differences in biochemical composition are preserved after drying and freezing.

Drying, a popular preservation method for storing raw plants, is considered a critical factor for postharvest management and the merchantability of aromatic herbs, including *Mentha* species. Depending on the drying method, the degree of influence of technological parameters on the biochemical composition of raw plants can vary [69,70]. In the drying process, because the chemicals in the cells are concentrated as a result of the evaporation of moisture, the osmotic pressure increases. Reducing the humidity level in greens from 75 to 80% to less than 15% during drying inhibits the growth of microbes and successfully preserves the raw material [71,72]. However, it is often accompanied by the loss of biologically active substances and their properties [71,73,74], which is consistent with the data obtained in our research. There are no data on the drying of different types of mint, but it was determined that the optimal temperature for *M. spicata* is 45 °C, as it retains the maximum number of chlorophylls and carotenoids and slightly increases the content of polyphenolic compounds [75,76].

When evaluating the quality of the dried mint greens, it should be noted that the amount of dry matter was concentrated during the drying process, and its level increased by 5.0–5.5 times. As a result, the sugar content of the dried mint greens increased by 4.9–5.2 times. But conversion to dry matter showed that the level of sugars during drying decreased by 10.3–13.0%. Spearmint (12.69 g $\times$ 100 g$^{-1}$) and peppermint (11.07 g $\times$ 100 g$^{-1}$) had the highest level of sugars. The total sugar content after freezing ranged from 5.8% in peppermint to 9.7% in spearmint.

Park et al. state that for carbohydrate preservation, optimal temperatures are 25 °C and 50 °C [77], emphasizing, however, that the preservation of metabolites depends strongly on the drying method. The drying method, therefore, has to be picked according to the optimum for the preservation of priority goal metabolites.

The changes in titratable acidity during drying were similar to the observed variations in sugar content: their level increased 3.4–4.2 times depending on the type of mint. However, in terms of dry matter, the content of titratable acidity decreased on average by 30%. The

titrated acidity of peppermint at freezing decreased by 7.3%; for spearmint, this indicator is 9.8%; and for horsemint and silver-leaved horsemint, it is 17.9 and 15.6, respectively.

Ascorbic acid is an unstable compound that can easily be degraded at elevated temperatures. During the drying of the raw material, it can reach 33% [52]. As observed in our data, the content of ascorbic acid in the dried mint greens ranged from 11.10 to 17.16 mg $\times$ 100 g$^{-1}$. The loss of this bio-antioxidant during drying was approximately 70% in terms of dry matter. The losses of ascorbic acid at freezing were significantly lower: from 3.2% in peppermint to 4.9% in spearmint.

Spearmint and peppermint (107.6 and 102.3 mg $\times$ 100 g$^{-1}$) had the highest number of carotenoids in dried greens. The loss of carotenoids during the drying of the raw plants was approximately 10–15%. Interestingly, the high impact of the drying methods and drying temperature on the carotenoid losses was also reported by Park et al., with the lowest losses observed in the freeze-dried samples [77]. Significantly lower losses of carotenoids (within 3%) were observed during freezing.

Chlorophylls are the most sensitive to heat, and their content decreased by 21 to 38% compared to the initial content in fresh greens, depending on the species of mint. Many authors have reported that frozen vegetables retain their color better due to reduced chlorophyll degradation [78]. A similar trend is observed in our study. Freezing mint destroys only approximately 5% of chlorophyll in all samples.

Fresh mint greens were characterized by high levels of phenolic compounds, the content of which decreased by 19–29% after drying (compared to dry matter). Freezing was reported to facilitate the extraction of some cell-bound compounds; namely, the total phenolic content of frozen mint was shown to increase slightly. This is possible due to the formation of ice crystals and rupturing of plant cells, which facilitates the easy penetration of the solvent [79]. Our results, on the contrary, show that the total content of phenolic compounds decreases during freezing. However, the number of losses is different for different types of mint. The lowest losses of phenolic substances at freezing were in horsemint and silver-leaved horsemint—5.1%. The largest losses of 8.4% were observed for peppermint.

## 5. Conclusions

As a result of the research, it was established that local mint species are characterized by a powerful complex of phytonutrients and can be used as ingredients of food products with functional properties.

When mint is dried, biologically active substances are concentrated, and such raw materials have advantages in terms of storage conditions and ease of use as an ingredient in food recipes. However, losses of ascorbic acid reach 70%; of carotenoids—approximately 10–15%; of chlorophylls—21 to 38%; and of phenolic substances—19–29%. Peppermint and spearmint had higher stability of biologically active compounds.

Frozen mint has limited use in the food industry. However, freezing preserves a greater amount of biologically active substances compared to dried mint. Therefore, the use of frozen green mint should be considered as a better alternative to dried products. It is necessary to expand the range of food products with the use of semi-finished frozen mint products. In addition, better stability of the chemical composition during freezing was demonstrated by the samples of horsemint and silver-leaved horsemint. Therefore, the purposeful selection of mint species and varieties for processing will allow us to preserve the maximum amount of biologically active substances and increase the content of phytonutrients in the finished products.

**Author Contributions:** Conceptualization, T.H. and O.P.; methodology, M.S. and L.K.; database creation, K.P. and W.B.; Software, M.J.; validation, K.F.-G.; formal analysis, M.K.; Supervision, T.H. All authors have read and agreed to the published version of the manuscript.

**Funding:** Financed from the subsidy of the Ministry of Education and Science for the Hugo Kołłątaj Agricultural University in Kraków for the year 2023.

**Institutional Review Board Statement:** Not applicable.

**Informed Consent Statement:** Not applicable.

**Data Availability Statement:** Not applicable.

**Conflicts of Interest:** The authors declare no conflict of interest.

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
