# Peer review of "Mint Plants (Mentha) as a Promising Source of Biologically Active Substances to Combat Hidden Hunger"

_sustainability, doi:10.3390/su151511648_

Round 1

Reviewer 1 Report

Present article entitled “Plants of the Genus Mentha (Mentha L.) as a Promising Source of Biologically Active Substances” covers a very interesting theme , However authors have carried out very basic experiment, during the study .

As the  authors mentioned biological active substances of   Mentha, but  authors did not performed any deep studies like, antimicrobial, antioxidant , total phenols,  and others biological tests. Authors should also run GCMS  to explored the  metabolites present in the  Mentha.

English of the article is very poor, need extensive revision

Present article entitled “Plants of the Genus Mentha (Mentha L.) as a Promising Source of Biologically Active Substances” covers a very interesting theme , However authors have carried out very basic experiment, during the study .

As the  authors mentioned biological active substances of   Mentha, but  authors did not performed any deep studies like, antimicrobial, antioxidant , total phenols,  and others biological tests. Authors should also run GCMS  to explored the  metabolites present in the  Mentha.

English of the article is very poor, need extensive revision

Author Response

Thank you for analyzing our manuscript! comments in the app. English language - correction made

Reviewer 2 Report

The topic covered in this paper is highly significant; however, its writing quality is lacking, evident in the presence of errors, awkward mistakes and excessively long and convoluted sentences. Furthermore, the paper is not well organized. Therefore, a comprehensive revision is necessary. it is imperative to conduct extensive editing to address issues with the English language.

Here below, a detailed list of the points the authors should amend:

-        The abstract needs to be rewritten as it lacks coherence.

-        The English language should be revised by a native speaker.

-        In the introduction, you should mention some biological molecules found in raw plants and their associated bioactivities.

-        The objective of the study is unclear.

-        The following sentence is too long: “The possibility of integrated use of plant components in food products to give them multifunctional properties as well as the pharmacological correction of alimentary diselementosis is of particular relevance”.

-        The introduction needs to be rewritten, avoiding too long sentences and unstructured ideas.

-        The species names of plants should be italicized (ensure consistency throughout the document).

-        You should add the coordinates of collection places.

-        Reorganize the material and methods section (each method should be written in its own paragraph)

-        How was it determined that the soil from which the materials were collected is not contaminated by heavy metals? Soil composition analysis should be conducted.

-        Specify whether the material used in the study was cultivated or wild-grown.

-        Based on my knowledge, essential oils in plants primarily serve to protect them under environmental conditions, so the idea that essential oil is more available in the morning is wrong. If it's the opposite, please provide a reference.

-        What is the reason behind measuring chlorophyll content when discussing nutritive value?

-        Results in Table 1 must be presented with statistical analysis.

-        Ensure uniformity of units with international system (replace mg/100g with mg. g-1).

-        What is the importance of measuring titratable acidity?

Ameliorate discussion section

Author Response

(The authors gave the same response as above.)

Reviewer 3 Report

1. Your manuscript references found wrong formatted, please revise the references according to SustainabilityFollow this format

Phaal, R.; Farrukh, C.J.P.; Probert, D.R. Technology roadmapping—A planning framework for evolution and revolution. Technol. Forecast. Soc. Change 200471, 5–26. 

2.  Remove the review literature section from your manuscript, there is no need in research paper

3. should write the objectives of this study in the end of Introduction section 

4. 3. Please read these three papers and add some valuable reasoning to justify your study findings, you can add in discussion section, hope it will support to your results

(i) Murtaza, G., Ahmed, Z., Eldin, S. M., Ali, B., Bawazeer, S., Usman, M., Iqbal, R., Neupane, D., Ullah, A., Khan, A., Hassan, M. U., Ali, I., & Tariq, A. (2023). Biochar-Soil-Plant interactions: A cross talk for sustainable agriculture under changing climate. Frontiers in Environmental Science, 11, [1059449]. https://doi.org/10.3389/fenvs.2023.1059449.

(ii) Murtaza, G., Ahmed, Z., Usman, M., Tariq, W., Ullah, Z., Shareef, M., ... & Ditta, A. (2021). Biochar induced modifications in soil properties and its impacts on crop growth and production. Journal of plant nutrition, 44(11), 1677-1691.

4. Materials and methodology section is too long, make it effective and short, its too long its many basic and unnecessary information given in this section, its boring for readers. 

5. Need to improve the abstract 

Author Response

(The authors gave the same response as above.)

Reviewer 4 Report

Dear authors, 

Thank you for your submission.

This manuscript shows a comparative study of the chemical composition of the plant genus Mentha. The major problem of the manuscript is the extract preparation and the need for chemical characterization. 

I would suggest defining the chemistry of these extracts with HPLC/MS to have a clear comparison and better comment on the basic chemical composition. 

The main issue is the dry matter of Mints are the highest dry mass compared to the fresh and frozen mint at the same weight (50 g). This may result in the highest chemical composition of dried mints, which is very biased. 

The purpose of this study is not a novelty and has little interest because the results are focused on the chemical compositions of the extract only. For these reasons, I strongly recommend to refuse this article for publication.

Author Response

(The authors gave the same response as above.)

Round 2

Reviewer 1 Report

Author  have significantly improved  the article. The article can now be accepted in the current form

Author  have significantly improved  the article. The article can now be accepted in the current form

Author Response

Thank you for the fruitful cooperation and positive evaluation!

Reviewer 2 Report

The manuscript has been developed; However, the authors didn't take my comment about statical analysis into consideratio. It can be considered for puplication after providing statical analysis.

Author Response

We appreciate your opinion, which is already reflected in our manuscript and will be in further research! We also want to disclose the entire block of research so that our results are clear!

Reviewer 3 Report

i recommend for publication, congratulation  

Author Response

(The authors gave the same response as above.)

Reviewer 4 Report

Dear Authors,

Thank you very much for submitting a revision. The new revision looks excellent and easy to follow than the previous MN.

However, there is some object 

to analyze the content of biologically active substances in the samples of mint of the most common species and its modification depending on the method of processing;

seem not completely done in this work. Maybe the authors should tone down a little bit.

I would reccommend minor revision, including discussion and conclusion was too short and mostly stick with the results only.